

# Notes on the cheek region of the Late Jurassic theropod dinosaur *Allosaurus*

Serjoscha W. Evers[1], Christian Foth[1] and Oliver W.M. Rauhut[2,3,4]

[1] Department of Geosciences, University of Fribourg, Fribourg, Switzerland
[2] Bayerische Staatssammlung für Paläntologie und Geologie, Staatliche Naturwissenschaftliche Sammlungen Bayerns (SNSB), München, Germany
[3] Department of Earth and Environmental Sciences, Palaeontology and Geobiology, Ludwig-Maximilians-Universität, München, Germany
[4] GeoBioCenter, Ludwig-Maximilians-Universität, München, Germany

## ABSTRACT

*Allosaurus*, from the Late Jurassic of North America and Europe, is a model taxon for Jurassic basal tetanuran theropod dinosaurs. It has achieved an almost iconic status due to its early discovery in the late, 19th century, and due to the abundance of material from the Morrison Formation of the western U.S.A., making *Allosaurus* one of the best-known theropod taxa. Despite this, various aspects of the cranial anatomy of *Allosaurus* are surprisingly poorly understood. Here, we discuss the osteology of the cheek region, comprised by the jugal, maxilla, and lacrimal. This region of the skull is of importance for *Allosaurus* taxonomy and phylogeny, particularly because *Allosaurus* has traditionally been reconstructed with an unusual cheek configuration, and because the European species *Allosaurus europaeus* has been said to be different from North American material in the configuration of these bones. Based on re-examination of articulated and disarticulated material from a number of repositories, we show that the jugal participates in the antorbital fenestra, contradicting the common interpretation. The jugal laterally overlies the lacrimal, and forms an extended antorbital fossa with this bone. Furthermore, we document previously unrecorded pneumatic features of the jugal of *Allosaurus*.

## INTRODUCTION

The theropod dinosaur *Allosaurus* is certainly one of the best-known dinosaur taxa for scientists and the general public alike. It was first described on the basis of a fragmentary specimen from the Late Jurassic Morrison Formation by *Marsh (1877)*. However, more complete material, including an almost complete skeleton from the same locality, Felch Quarry, as the type and several skulls from other Morrison localities were referred to the same taxon shortly after (*Marsh, 1884*; *Osborn, 1903*, *1912*). The former specimen was described in detail in a monograph by Charles Gilmore in 1920 (although under the name *Antrodemus*; see *Madsen (1976)* for discussion), through which it became a reference taxon for theropod anatomy in general.

Corresponding author
Serjoscha W. Evers,
serjoscha.evers@googlemail.com

A large assemblage of theropod bones was found in sediments of the Morrison Formation close to Cleveland, Utah, in 1927, and excavation at the Cleveland Lloyd Dinosaur Quarry in subsequent decades has yielded a vast amount of Late Jurassic dinosaur specimens (see *Madsen, 1976*; *Gates, 2005*; *Peterson et al., 2017*). The most common dinosaur found at that site is *Allosaurus*, which is represented by at least 46 individuals (*Carpenter, 2010*), although the material is generally found disarticulated. The availability of such a large number of specimens of a single taxon led *Madsen (1976)* to publish a revised osteology of *Allosaurus*, in which he figured every individual bone for this genus, often in several views. It should be noted here that *Madsen (1976*: 2*)* himself noted that his description and illustrations were not intended to give an accurate account of the morphology of any individual element, but rather provide a composite reconstruction of the anatomy of this taxon. Nevertheless, due to his work, *Allosaurus* has become one of the best and most completely known theropod taxa, which is widely used in studies of theropod phylogeny, geometric morphometrics, biomechanics, and biology in general (*Gauthier, 1986*; *Holtz, 1994*; *Rogers, 1998*, *2005*; *Hanna, 2002*; *Rauhut, 2003*; *Rayfield et al., 2001*; *Rayfield, 2005*; *Carrano, Benson & Sampson, 2012*; *Brusatte et al., 2012*; *Foth & Rauhut, 2013*; *Snively et al., 2013*; *Lautenschlager, 2015*; *Foth et al., 2015*).

Due to the large number of specimens known for *Allosaurus*, several authors have observed variation among the material (*Chure & Madsen, 1996*; *Smith, 1998*; *Chure, 2000*; *Carpenter, 2010*; *Loewen, 2009*), arriving at different conclusions regarding the taxonomy of the genus *Allosaurus*. Because the holotype material of the type species *A. fragilis* is not diagnostic, USNM 4734, the nearly complete specimen from Felch Quarry (*Gilmore, 1920*; *Carrano, Loewen & Evers, 2018*), was proposed as a neotype (*Paul & Carpenter, 2010*; supported by several comments, for example, *Carpenter & Paul, 2015*; *Carrano, Loewen & Evers, 2018*). A second North American species, originally informally diagnosed in an unpublished PhD thesis (*Chure, 2000*) has recently been formally named as *Allosaurus jimmadseni* (*Chure & Loewen, 2020*). Further putative species, *A. lucasi* and *A. amplus*, are based on very fragmentary and probably undiagnostic material (*Dalman, 2014*; *Galton, Carpenter & Dalman, 2015*). As the Morrison Formation was deposited over a duration of 7 million years and crops out over 1.2 million km$^2$ (*Maidment & Muxworthy, 2019*), reported variation among specimens of *Allosaurus* could possibly be explained by geographic or stratigraphic separation of occurrences. The taxonomy of *Allosaurus* needs to be revised, but this should only be done when the neotype for *A. fragilis* has been formalised by an ICZN decision, so it can be compared in detail with the species described by *Chure & Loewen (2020)*. Here, we use the taxon *Allosaurus* without species epithet due to the unsolved taxonomic issues. However, our observations are based on specimens that have been referred to both species, and we have not found any differences between those for the elements of interest.

The cranial morphology of *Allosaurus* was first described by *Osborn (1903*, *1912)* and *Gilmore (1920)*. These descriptions were based on three almost complete, but partially disarticulated and/or distorted and damaged skulls, two from Bone Cabin Quarry (*Osborn, 1903*, *1912*) and one from the type locality of the genus, Felch Quarry (*Gilmore, 1920*). All specimens were, unfortunately, damaged or incomplete in the anterior cheek region,

and although both *Osborn (1903*: 697*)* and *Gilmore (1920*: 29*)* stated that the jugal formed part of the margin of the antorbital fenestra, this was not unambiguously clear from their illustrations, as parts of this region were reconstructed.

In contrast, *Madsen (1976*: pl. 1*)* reconstructed the skull of *Allosaurus* with an anteriorly tapering jugal that is excluded from the margin of the antorbital fenestra in lateral view. This reconstruction turned out to be very influential, with consequences for several kinds of studies including this taxon. Thus, in a multitude of phylogenetic studies that used differences in the expression of the jugal on the rim of the antorbital fenestra as a phylogenetic character, *Allosaurus* was scored as lacking such an expression (*Holtz, 1994*, *1998*; *Currie & Carpenter, 2000*; *Rauhut, 2003*; *Holtz, Molnar & Currie, 2004*; *Smith et al., 2007*; *Benson, Carrano & Brusatte, 2010*; *Carrano, Benson & Sampson, 2012*). Besides, a study of the biomechanical significance of suture morphology of this taxon also used this configuration (*Rayfield, 2005*). Furthermore, the clear presence of an expression of the jugal on the rim of the antorbital fenestra was considered an important character to distinguish the European species of *Allosaurus*, *A. europaeus*, from its North American counterparts (*Mateus, Walen & Antunes, 2006*; see also *Malafaia et al., 2007*).

Here, we review the evidence for the configuration of the maxilla, lacrimal and jugal and its significance for the question whether the latter bone participated in the rim of the antorbital fenestra in *Allosaurus*.

## MATERIALS AND METHODS

In order to assess the configuration of the anterior cheek region of *Allosaurus*, we studied articulated skulls (DINO 11541; MOR 693; DINO 2560 (UUVP 6000)), a disarticulated skull (SMA 0005), and isolated elements of this taxon from the Morrison Formation of North America. Isolated elements included numerous specimens of maxillae, jugals and lacrimals from the Cleveland-Lloyd Dinosaur Quarry of Utah, from which several elements were selected, in which the regions of interest are particularly well preserved. These specimens included three left maxillae (UMNH VP 9168, 9208 and 9216), a left (UMNH VP 9475) and a right lacrimal (UMNH VP 9473), and two right (UMNH VP 9083 and 9085) and one left jugal (UMNH VP 8972). Two further left jugals (UMNH VP 8973 and 8974) were documented, because in these pneumatic features were well visible due to breakage.

## RESULTS

### The configuration of the anterior cheek in *Allosaurus*: Madsen's interpretation

As noted above, *Madsen (1976)* described the osteology of *Allosaurus* on the basis of abundant, but disarticulated material from the Cleveland-Lloyd dinosaur quarry of Utah, although he used a partially articulated specimen from Dinosaur National Monument, DINO 2560 (formerly UUVP 6000), as guidance (*Madsen, 1976*: 2). In his skull reconstruction, *Madsen (1976*: pl. 1*)* illustrated a broad contact between the ventral process of the lacrimal and the posterior process of the maxilla, visible in lateral view. Both bones form the posteroventral margin of the internal antorbital fenestra, while the jugal is

excluded from the antorbital fenestra. In contrast to the individual reconstruction of the jugal (*Madsen, 1976*: pl. 4D, E), the anterior process of the jugal in the skull reconstruction was illustrated to be subdivided into a long and tapering anteroventral and a shorter posterodorsal process, which together formed a deeply concave anterodorsal margin. In his figures of the individual elements, *Madsen (1976)* correctly illustrated the jugal with a pronounced anterior expansion, but indicated that most of this expansion would have been overlapped laterally by the lacrimal in the articulated skull (*Madsen, 1976*: pl. 4D), thus interpreting the depressed area on the anterior expansion as the facet for the latter bone. His interpretation was probably influenced by the curved rim of the antorbital fossa on the jugal, which stands out prominently in articulated skulls, and was interpreted as the jugal-lacrimal suture, and the very thin bone anterior to it, which resembles the distal end of the ventral process of the lacrimal.

## Configuration of the anterior cheek in other theropods

The morphology of the cheek region of theropod dinosaurs has recently been reviewed by *Sullivan & Xu (2017)* and *Wang & Hu (2017)*, focusing primarily on the morphology of the jugal. Apart from a few exceptions, the anterior process of the jugal in theropods participates in the posteroventral margin of the antorbital fenestra. In small-bodied theropods this process is usually slender and tapering, but it is dorsoventrally expanded in many large-bodied taxa.

In contrast, the exclusion of the jugal from the antorbital fenestra is occasionally present in theropods, including various coelophysids (*Raath, 1977*; *Colbert, 1989*; *Rowe, 1989*; *Tykoski, 1998*; *Bristowe & Raath, 2004*), the ceratosaurid *Ceratosaurus* (*Gilmore, 1920*; *Madsen & Welles, 2000*), and the basal alvarezsaurid *Haplocheirus* (*Choiniere et al., 2014*), while it is the common morphology in non-avian Pygostylia (*Wang & Hu, 2017*). In addition, the configuration was described for the basal theropod *Zupaysaurus* (*Ezcurra, 2007*) and the megalosaurid *Torvosaurus* (*Brusatte et al., 2010*). However, further preparation of the anterior cheek region of *Zupaysaurus* revealed a jugal contribution to the antorbital fenestra (M. Ezcurra, 2012, personal communication), while the incomplete preservation of the maxilla and jugal in *Torvosaurus* does not allow a proper judgement of the true morphology. However, all taxa for which the exclusion of the jugal from the antorbital fenestra can be confirmed with no doubt show a laterally exposed contact between maxilla and lacrimal, the extent of which depends primarily on the shape of the lacrimal ventral process. Accordingly, the contact is very broad in *Coelophysis* and *Ceratosaurus*.

Regardless of the jugal contribution to the margin of the antorbital fenestra, the relative arrangement and articular surfaces of bones involved in the formation of the cheek are the same in all non-avian theropods: the jugal overlaps the lateral surface of the ventral process of the lacrimal. As noted by *Sereno & Novas (1993)*, this is a saurischian synapomorphy. Consequentially, the ventral end of the lacrimal is positioned medially to the jugal, so that a lacrimal-maxilla contact is not externally visible in taxa in which the jugal extends to the antorbital fenestra. However, even in taxa with this configuration, there is an internal contact between the lacrimal and the maxilla. The usually

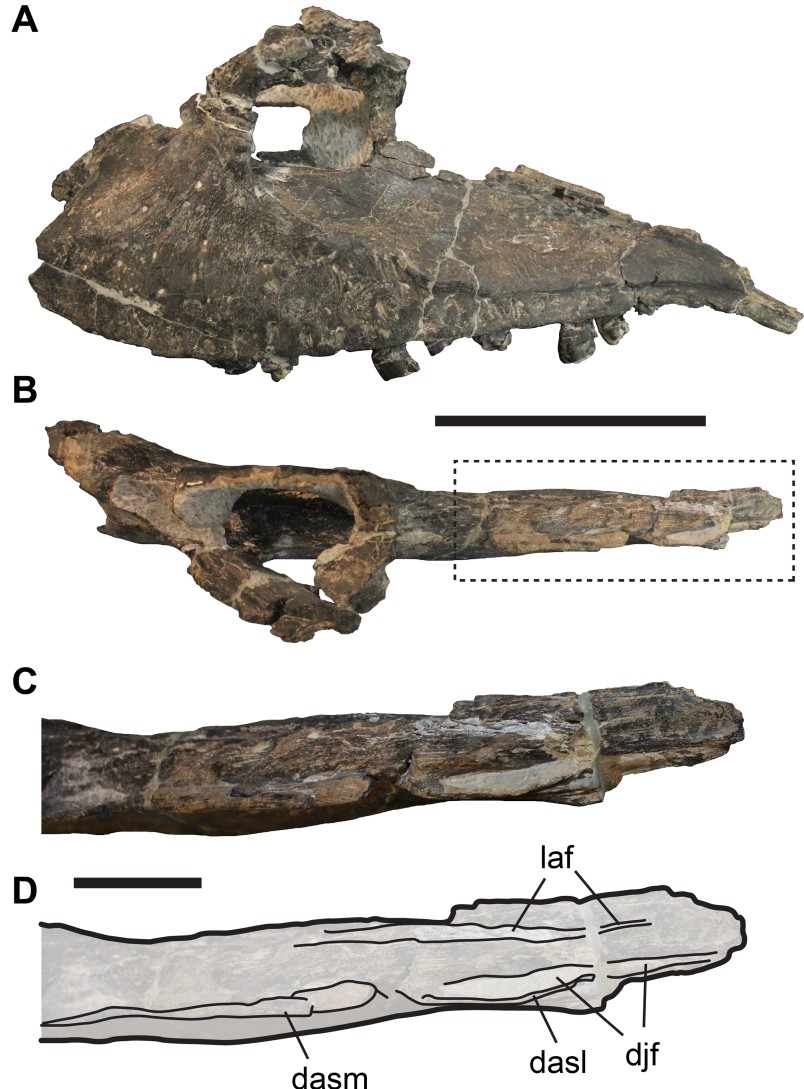

**Figure 1** **Incompletely preserved left maxilla of UMNH VP 9216,** *Allosaurus fragilis,* **showing details of the posterior process.** (A) Lateral view. (B) Dorsal view. (C) Close-up of posterior process in dorsal view. (D) Line-drawing of C. Dashed box in (B) shows region shown in more detail in (C) and (D). Abbreviations: dasl, dorsally ascending lamina; dasm, dorsally ascending margin of posterior process; djf, dorsal jugal facet of maxilla; laf, lacrimal facet. Scale bar in (A and B) equals 10 cm; scale bar in (C and D) equals three cm.

anteroposteriorly expanded basal plate of the lacrimal sits in a facet on the dorsal shelf of the maxilla that is situated medially to the groove for the jugal. This is the case even in taxa in which the lacrimal seems to be dorsoventrally short and is widely separated from the maxilla in external view of the articulated skull, such as in *Herrerasaurus* (PVSJ 53).

## Data from specimens of *Allosaurus*

The posterior end of the maxilla of *Allosaurus* shows facets for the articulation with the jugal, lacrimal and palatine, which are roughly mediolaterally aligned. The contact with the jugal is positioned laterally with regard to the contact with the lacrimal, and both these

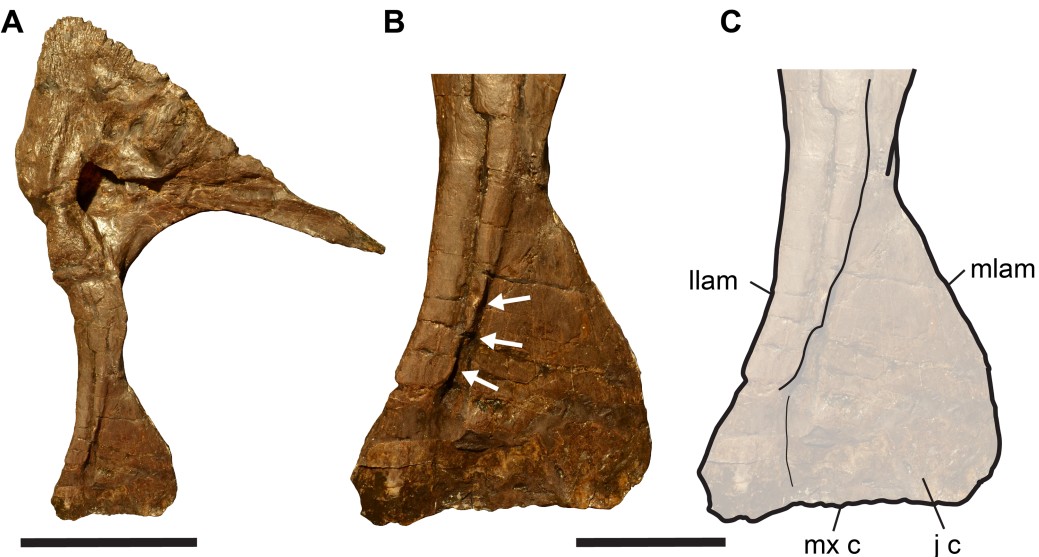

**Figure 2 Completely preserved right lacrimal of SMA 0005, *Allosaurus jimmadsoni*.** (A) Lateral view. (B) Close-up of ventral process in lateral view. (C) Line-drawing of B. Arrows in (B) indicate groove for articulation of jugal. Abbreviations: j c, jugal contact; llam, lateral lamina; mlam, medial lamina; mx c, maxilla contact. Scale bar in (A) equals 10 cm; scale bar in (B and C) equal three cm.

contact facets form grooves on the dorsal surface of the posterior processes of the maxilla (Fig. 1). The facet for the palatine is the medialmost of the three contacts, and is visible on the medial surface of the maxilla.

The jugal facet is developed as a narrow, dorsally facing groove (Figs. 1C and 1D), which extends from the posteroventral corner of the bone to the level of the third alveolous as counted from posterior. The posteriormost part of this groove is exposed laterally, but a dorsally ascending lamina conceals the anterior part of the groove in lateral view (Figs. 1C and 1D). The lacrimal facet is subparallel to the jugal facet, but separated from the former by a low, but relatively broad ridge (Figs. 1C and 1D). The lacrimal facet itself is developed as a subtle groove, which extends along the medial margin of the dorsal surface of the posterior process of the maxilla. This facet continues marginally further anteriorly than the facet for the jugal, forming a broad contact between maxilla and lacrimal. The third articulation facet, the palatine facet of the maxilla, is much broader than the other facets described above. It is positioned medial to the lacrimal contact, and is developed as a roughened longitudinal area that spans from the first to approximately the seventh tooth position as counted from posterior. The dorsal margin of the palatine facet is developed as a near vertical shelf of bone, which prohibits a contact between the palatine and lacrimal.

The lacrimal has a mediolaterally thin, and anteroposteriorly expanded ventral process that articulates with the maxilla and jugal (Fig. 2). The ventral process can be divided into two units. Anteriorly and ventrally, the ventral process forms a thin blade of bone (medial lamina), which is recessed from a thickened posterior margin (lateral lamina) (Figs. 2B and 2C). A vertically directed, anteriorly facing groove invades the thick posterior

margin at the posterior end of the thin blade (Fig. 2B). We interpret this incision as a facet for the posterior margin of the anterior blade of the jugal. Consequently, the anterior process of the jugal covers large parts of the lacrimal blade laterally when both bones are articulated. In his reconstruction of the lacrimal, Madsen (1976: pl. 5A) illustrated a deep notch in the ventral margin of lacrimal. However, as this region is often broken in Allosaurus specimens (see Osborn, 1903; Carpenter, 2010), the presence of such a notch is probably an artefact. In those specimens (e.g. SMA 0005) in which the ventral end of the ventral process is fully intact, this margin is almost straight (Fig. 2). This observation fits with the dorsally exposed lacrimal facet groove of the maxilla.

The jugal of Allosaurus has a dorsally expanded anterior process that contacts the maxilla and lacrimal. This process is often incompletely preserved (even in articulated specimens), but it is nearly completely preserved in the specimen SMA 0005 (Fig. 3). The jugal of Allosaurus is relatively tightly articulated with the maxilla via a ventral and a medial contact. The ventral contact is formed by the relatively thin, keel-like margin of the jugal, which slots into the dorsally exposed jugal facet on the posterior process of the maxilla. The second facet is a wedge-shaped, posteriorly tapering depression in the lateral surface of the jugal, which receives the lateral part of the posterior process of the maxilla (Fig. 3).

The lateral surface of the anterior process of the jugal is characterised by a sharp, concavely curved step-like ridge, which separates the process into an extremely thin, blade-like anterodorsal region, which is recessed from a thicker posteroventral region (Fig. 3). We identify this ridge as the posteroventral margin of the antorbital fossa. This margin is slightly excavated to a shallow groove posteroventrally, as evident from several better-preserved specimens, such as UMNH VP 9085, UMNH VP 8972 and SMA 0005. Unlike reported in other works (Brusatte et al., 2010; Eddy & Clarke, 2011), there is a small pneumatic foramen located within the margin of this groove (see Currie & Zhao, 1993; Coria & Currie, 2006). The foramen excavates posteriorly into the anterior process of the jugal (Fig. 4). Evidence for the pneumatic invasion of the jugal via the anterior process is also given by several specimens in which the anterior process of the jugal is broken off, exposing a pneumatic recess within it (e.g. UMNH VP 8973, UMNH VP 8974; Fig. 4). Because the anterior blade is extremely thin, it is often incompletely preserved (see Chure, 2000; Loewen, 2009; Carpenter, 2010), leading to different interpretations regarding the anterodorsal morphology of the process, specifically with regard to its extend into the antorbital fenestra (e.g. Madsen, 1976 vs. this study). However, some specimens (e.g. SMA 0005) show that the anterodorsal margin is convexly rounded, as reconstructed by Madsen for the isolated jugal (1976: pl. 4D, E). The thickened posterior margin of the anterior jugal process faces towards the orbit and slots into the facet in the lateral lamina of the lacrimal (see above). Consequentially, the lacrimal wraps around the posterior edge of the jugal, which is particularly well visible in articulated specimens (Fig. 5). The same articulation is also present in Acrocanthosaurus (NCSM 14345, S.W. Evers, C. Foth & O.W.M. Rauhut, 2012, personal observation). This contact appears to be relatively tight, so that kinematic movements between the lacrimal and jugal seem unlikely.
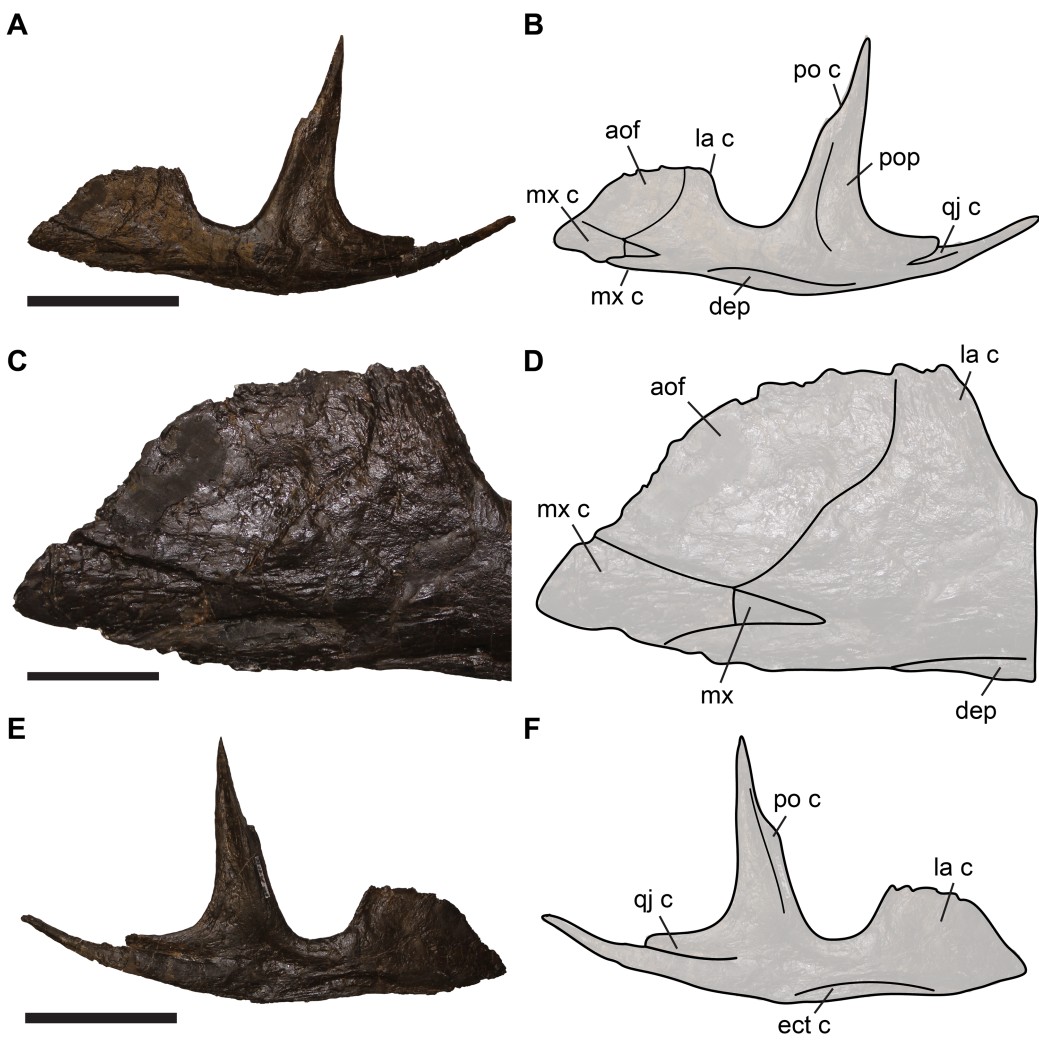

**Figure 3 Completely preserved left jugal of SMA 0005, *Allosaurus jimmadsoni*.** (A) Lateral view. (B) Line drawing of A. (C) close-up of anterior jugal process in lateral view. (D) Line-drawing of B. (E) Medial view. (F) Line-drawing of E. Abbreviations: aof, antorbital fossa; dep, depression; ect c, ectopterygoid contact; la c lacrimal contact; mx, maxilla; mx c, maxilla contact; po c, postorbital contact; pop, postorbital process of jugal; qj c, quadratojugal contact. Scale bars in A–B, E–F equal 2 cm; scale bar in C–D equals 3 cm.

The thin jugal blade lies on the lateral surface of the medial lamina of the lacrimal. The low ridge that marks the margin of the antorbital fossa on the jugal aligns with the edge of the posteriorly thickened margin of the lacrimal, so that the antorbital fossa is continuous between both bones. This morphology can be also observed in various articulated *Allosaurus* skulls, including MOR 693 (S.W. Evers, 2014, personal observation), UUVP 6000 (S.W. Evers & O.W.M. Rauhut, 2016, personal observation), and DINO 11541 (S.W. Evers & O.W.M. Rauhut, 2016, personal observation) (Fig. 3).

## DISCUSSION

The re-examination of the bones of the anterior cheek region in *Allosaurus* demonstrates that the famous skull reconstruction by *Madsen (1976)* is erroneous with respect to

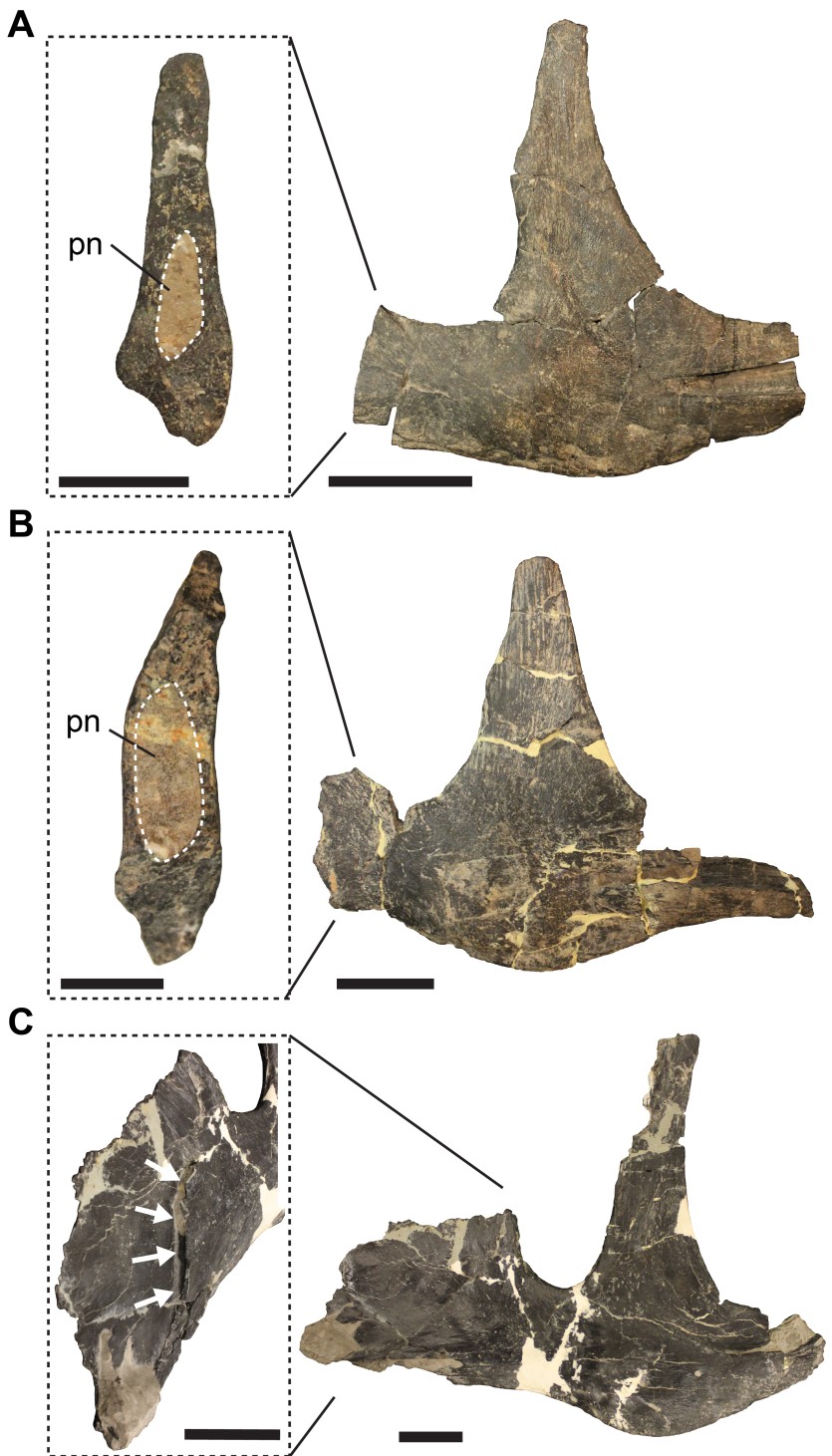

**Figure 4** **Jugal pneumatisation in *Allosaurus fragilis*.** (A) Left jugal UMNH VP 8973 in lateral view and with close-up on broken anterior process, revealing pneumatic recess. (B) UMNH VP 8974 in lateral view and with close-up on broken anterior process, revealing pneumatic recess. (C) Right jugal UMNH VP 9085 in lateral view and anterolateral close-up of anterior process, showing pneumatic opening in the margin of the antorbital fossa. Note that images in (C) are reflected for comparison. Abbreviations: pn, pneumatic recess. Scale bars in close-ups equal one cm, scale bars for lateral views equal three cm.

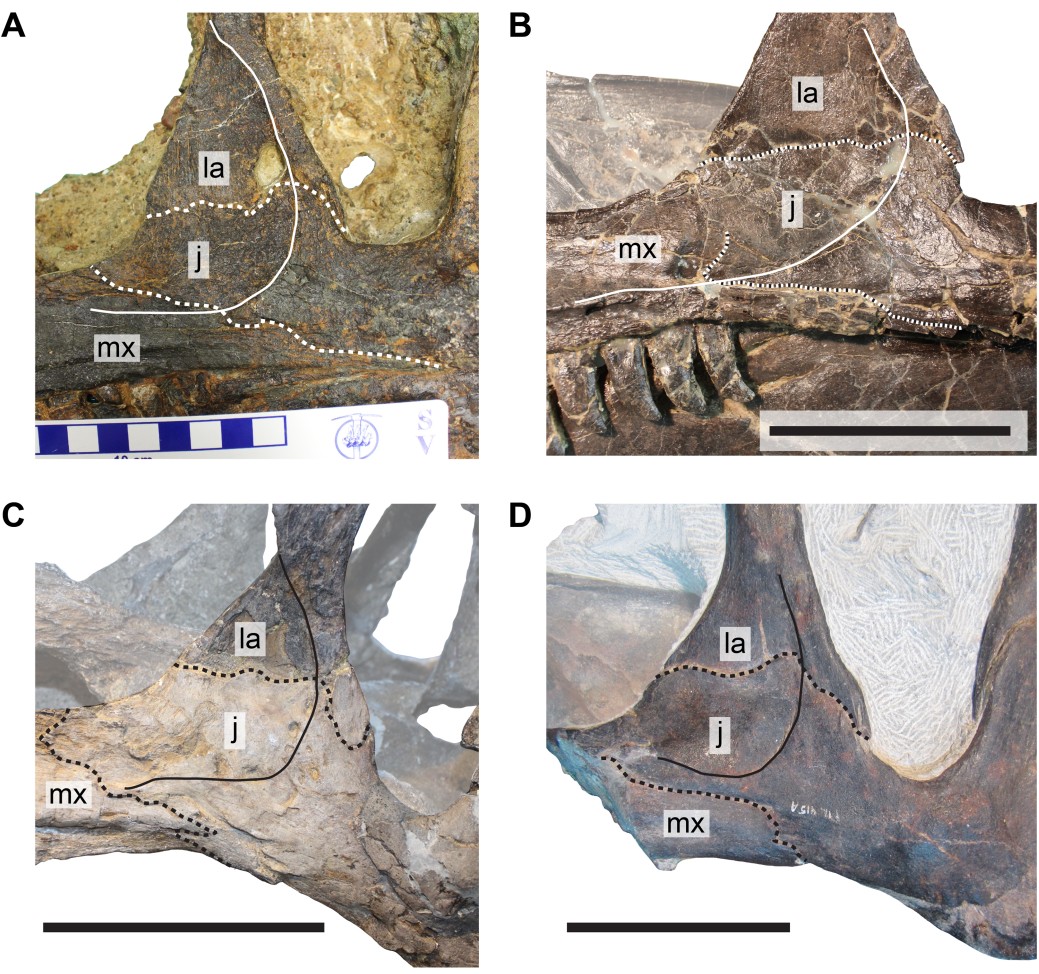

**Figure 5 Comparison of cheek regions in different specimens of *Allosaurus*.** (A) Left cheek region of DINO 11541, designated holotype of *A. jimmadseni*. (B) Left cheek region of MOR 693, *A. jimmadseni*. (C) Reflected right cheek region of DINO 2560 (formerly UUVP 6000), *A. fragilis*. (D) Left cheek region of ML 415, holotype of *A. europaeus*. Abbreviations: j, jugal; la, lacrimal; mx, maxilla. Dashed lines represent bone sutures discussed in the text, and full lines represent the posteroventral margin of the antrobital fossa. Scale bars in (B–D) equal 10 cm, squares on scale bar in (A) each equal one cm.

morphology of the anterior process of the jugal and its articulation with the lacrimal and maxilla. The anterior process of the jugal in *Allosaurus* is in fact enlarged and plate-like (Fig. 3) and covers the lateral side of the lacrimal in its ventral part (Figs. 5 and 6). The anterodorsal margin of the anterior process of the jugal extends into the internal antorbital fenestra. This morphology was previously described by *Osborn (1903)* for the disarticulated specimens AMNH 600, and by *Gilmore (1920)* for the artificially articulated USNM 4734. In addition, other skull reconstructions based on UUVP 6000 were illustrated with this configuration too (see *Bakker, 1998*: fig. 3B; *Paul, 2002*: fig. 10.2F; *Fastovsky & Weishampel, 2005*: fig. 12.2F), but without commenting on the discrepancy to *Madsen's (1976)* reconstruction of the same specimen. As *Madsen (1976*: pl. 4D, E) figures the morphology of the anterior process of the jugal correctly in the individual bone
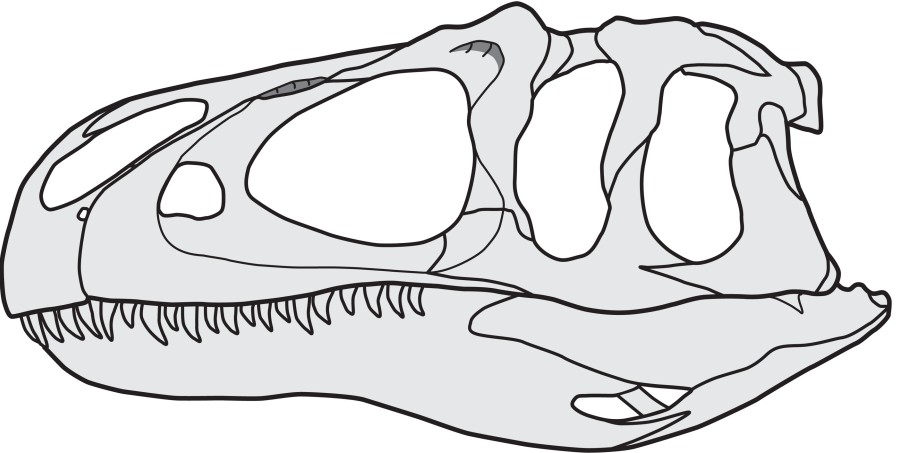

**Figure 6 Reconstruction of the skull of *Allosaurus*, based on MOR 693 (*A. jimmadseni*).** Note that the jugal participates in the antorbital fenestra, and that the lacrimal overlaps the posterior margin of the anterior jugal process.

reconstructions, we can only speculate why his reconstruction of the skull is erroneous. Based on its position, the concavely shaped and gently recessed anterodorsally surface of the anterior process (*Madsen, 1976*: pl. 1) clearly represents the jugal part of the antorbital fossa, which is continuous with the respective margins of the ventral process of the lacrimal and posterior process of the maxilla.

However, our current observations confirm a broad contact between maxilla and lacrimal in *Allosaurus* as illustrated in *Madsen (1976*: pl. 1*)*, but the articulation is covered laterally by the anterior process of the jugal and only visible from medial view. A similar morphology can be found in the carcharodontosaurid *Acrocanthosaurus* (right side of NCSM 14345, S.W. Evers, C. Foth & O.W.M. Rauhut, 2012, personal observation). In addition, *Hendrickx & Mateus (2014)* described a prominent medially located articulation facet for the lacrimal on the dorsal side of the distal end of the posterior process of the maxilla of *Torvosaurus gurneyi*. This contact is also present in ornithomimosaurs (*Kobayashi & Lü, 2003*), therizinosaurids (*Clark, Perle & Norell, 1994*; *Lautenschlager et al., 2014*), oviraptorosaurs (*Clark, Norell & Rowe, 2002*; *Balanoff et al., 2009*; *Balanoff & Norell, 2012*) and non-avian Pygostylia (*Wang & Hu, 2017*), while it is absent in abelisaurids (*Bonaparte, Novas & Coria, 1990*; *Sampson & Witmer, 2007*; *Canale et al., 2009*), tyrannosaurids (*Currie, 2003*), dromaeosaurids (*Turner, Makovicky & Norell, 2012*), troodontids (*Tsuihiji et al., 2014*) and crown-group birds (*Zusi, 1993*).

Furthermore, the new observations have implications for the diagnosis of the European *A. europaeus* (*Mateus, Walen & Antunes, 2006*; Fig. 5D) as a distinct species, for which the jugal participation in the antorbital fenestra was listed as one of the few autapomorphic characters that differentiate it from the North American species. Besides, the authors listed the absence of a lacrimal-maxilla contact as a further apomorphy, which is related to the former character. However, as pointed out above, this is only true for the lateral view, while a medial contact between both bones was almost certainly present. Unfortunately, this cannot be verified at the moment as the internal side of the skull is

filled with matrix, but the consistent nature of this contact in regard to *Allosaurus* specimens examined for this study allow inferring the presence of this contact with high confidence. All other diagnostic features of *A. europaeus* have been questioned to be truly unique, and some have proven to be variably present in North American *Allosaurus* specimens (*Malafaia et al., 2007*). Therefore, a re-evaluation of the European species is necessary, as currently none of the originally proposed diagnostic features are uniquely present in the holotype of *Allosaurus europeaus*.

## CONCLUSIONS

The cheek region of *Allosaurus* conforms to the general pattern observed in basal tetanurans: the jugal overlies the lateral surface of the lacrimal, and both bones articulate with the maxilla. The anterior process of the jugal of *Allosaurus* is anterodorsally expanded and contributes to the antorbital fenestra and forms parts of the antorbital fossa, contradicting the famous reconstruction by *Madsen (1976)*. The articulation facets between the maxilla, lacrimal and jugal are relatively complex and indicate that the contacts between these cheek bones were relatively strong, probably allowing little if any movement. The configuration of cheek bones does not vary between the examined specimens in *Allosaurus*, and our observations furthermore indicate that the European species *A. europaeus* did not differ in this regard from North American material.

## INSTITUTIONAL ABBREVIATIONS

| | |
|---|---|
| **AMNH** | American Museum of Natural History, New York, USA |
| **DINO** | Dinosaur National Monument, Jensen, Utah, USA |
| **ML** | Museu da Lourinhã, Lourinhã, Portugal |
| **MOR** | Museum of the Rockies, Bozeman, Montana, USA |
| **NCSM** | North Carolina Museum of Natural Sciences, Raleigh, North Carolina, USA |
| **PVSJ** | Paleontología de Vertebrados, Universidd de San Juan, Argentina |
| **SMA** | Saurier-Museum Aathal, Switzerland |
| **UMNH** | Utah Museum of Natural History, Salt Lake City, Utah |
| **USNM** | United States National Museum of Natural History, Washington DC, USA. |

## ACKNOWLEDGEMENTS

The authors would like to thank several people who provided access to specimens under their care. These are, in no particular order, Carrie Levitt-Bussian and Randall Irmis (UMNH), Brooks Britt and Rodney Scheets (BYU), Dan Chure (DINO/DNM), Octavio Mateus and Simão Mateus (ML), Vince Schneider and Lindsay Zanno (NCSM), Jack Horner, John Scanella and Bob Harmon (MOR), Kirby Siber, Thomas Bollinger and Ben Pabst (SMA), Mark Norell and Carl Mehling (AMNH), Paul Barrett (NHMUK). We would also like to thank Roger Benson for providing additional photographs of *Neovenator salerii*. We are thankful to Mark Loewen, Matt Carrano, Dan Chure, and Octavio Mateus for numerous discussions about *Allosaurus*. We are thankful for insightful reviews by Elisabete Malafaia and Thomas Holtz, which improved a previous version of this work.

## Funding

This work was supported by the Swiss National Science Foundation (PZ00P2_174040) awarded to Christian Foth. Oliver Rauhut was supported by a Volkswagen Foundation (I/84 640) grant. The funders had no role in study design, data collection and analysis, decision to publish, or preparation of the manuscript.

## Grant Disclosures

The following grant information was disclosed by the authors:
Swiss National Science Foundation: PZ00P2_174040.
Volkswagen Foundation: I/84 640.

## Competing Interests

The authors declare that they have no competing interests.

## Author Contributions

- Serjoscha W. Evers conceived and designed the experiments, performed the experiments, analysed the data, prepared figures and/or tables, authored or reviewed drafts of the paper, and approved the final draft.
- Christian Foth performed the experiments, analysed the data, authored or reviewed drafts of the paper, and approved the final draft.
- Oliver W.M. Rauhut conceived and designed the experiments, performed the experiments, analysed the data, authored or reviewed drafts of the paper, and approved the final draft.

## Data Availability

The specimens used for conducting this study are available at their museum repositories:

AMNH 600; DINO 11541; DINO 2560; ML 415; MOR 693; NCSM 14345; PVSJ 53; SMA 0005; UMNH VP 8972, 8973, 8974, 9083, 9085, 9168, 9208, 9216, 9473, 9475; USNM 4734.

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
