# Peer review of "Notes on the cheek region of the Late Jurassic theropod dinosaur Allosaurus"

_PeerJ, doi:10.7717/peerj.8493_

## Round 0.1 · original submission · Minor Revisions

Dear authors,

I have accepted the decision of ‘minor revisions’ from the reviewers.

Please take note of reviewer 2’s comments regarding the usage of an unpublished Allosaurus name. Please do not use a name that is not ‘published’ under the Articles of the ICZN Code.

I look forward to receiving your revised manuscript.

·

Basic reporting

The authors describe several interesting features of the well-known theropod dinosaur Allosaurus and analyze some characters that have been used on phylogenetic analyses of this taxon. The intraspecific variability of this taxon is a topic that has been broadly discussed. This work adds important data to the knowledge of the cranial anatomy of Allosaurus as well as to the taxonomic and phylogenetic interpretation of some cranial features, especially related with the European species Allosaurus europaeus.
The manuscript is clearly written in professional, unambiguous language, the literature references are updated, the context is well described.

Experimental design

This is an original research, the questions are well defined, relevant and meaningful, the investigation is rigorous and the methods are described with sufficient detail.

Validity of the findings

The figures are high quality, well labeled, and described, and the conclusions are relevant and well stated.

Additional comments

I made some notes directly on the manuscript. Apart from these notes, I have only few additional comments and suggestions:
- On page 7, line 70, I think you should include the references of Yun. (2019), Carrano, Loewen, and Evers. (2018), and Carpenter and Paul (2015) to the case.
- On page 8, the phrase on lines 90 – 95 is too large, I suggest splitting it in two.
- Along the manuscript you use Allosaurus “jimmadsoni”, but the name proposed by Chure (2000) is Allosaurus “jimmadseni”.

Apart from this, I strongly recommend the publication of this manuscript.

·

Basic reporting

I find no particular faults with the structure of the paper, the quality of the text, or the soundness of the arguments.

The relevant literature is extensively cited.

The figures provided are necessary and sufficient to make their case.

Experimental design

Their observations are sufficiently well documented to be replicable by other workers. Indeed, by using high-quality photographs rather than interpretative drawings they will help to rectify the particular problems concerning the interpretation of the anatomy of the jugal that this paper is meant to address.

Validity of the findings

The conclusions of the authors are sound, and will aid in future studies of theropod dinosaur anatomy and taxonomy.

Additional comments

The authors should be congratulated on this brief review of just one relatively small section of the skull that nevertheless has important implications for theropod taxonomy and systematics. Their work helps to correct a very, very widespread anatomical misinterpretation (they note a number of different researchers and studies which used this misinterpretation). Their observations are strongly supported and documented.

I see very little to recommend in changing for a published version of this. There are a couple of typographic errors that I noted. One is spread throughtout the text, and the other is a minor one.

The first of these concerns the as-yet-not-formally-named second North American species of Allosaurus. The proper spelling of the forthcoming new Allosaurus species is “jimmadseni”. That said, even though this is the worst-kept secret in theropod taxonomy, I think it might be better to not refer to this forthcoming taxon by name. I have heard that Chure is working with additional colleagues to finally get this name published, but I do not know the status of that paper. So it might be better to refer to it as “Allosaurus Morrison sp. 2” or something of that nature. Invertebrate paleontologists and micropaleontologists seem comfortable in referring to unnamed “sp. 2” and the like until the formal names are applied, so it might be useful for vertebrate paleontologists to do so as well.

Line 110: A typographic error: “Ceveland” for “Cleveland”

The observations documented here will be very helpful in future studies, both with regards to the external antorbital fossa configuration and the internal pneumatic structures.

I am intrigued by the possibility that A. europaeus might not be separable from one or the other North American species.

Reviewer 3 ·

Basic reporting

The article is well written, clear and based on useful observations, and references. The revisions are very simple and straight forward:
L.72 and figure captions: The spelling "jimmadsoni" should be "jimmadseni" after the Jim Madsen and according to Chure (2000: p. 22). 
L. 118: Missing "ML"
L. 171, 304 and others: please use lower case after colon punctuation (:).
L. 476: space "from the"Figure 6 captions: clarify the Allosaurus species.
It would be interesting to hear comments about the validity of A. jimmadseni, A. atrox, A. amplus, A. lucasi  and contributions to the diagnosis (synapomorphies) of the genus and each one of the species.

Experimental design

The MS is based in specimen observations, that seem accurated.

Validity of the findings

The results seem valid.

Additional comments

The article is well written, clear and based on useful observations, and references. The revisions are very simple and straight forward:
L.72 and figure captions: The spelling "jimmadsoni" should be "jimmadseni" after the Jim Madsen and according to Chure (2000: p. 22). 
L. 118: Missing "ML"
L. 171, 304 and others: please use lower case after colon punctuation (:).
L. 476: space "from the"Figure 6 captions: clarify the Allosaurus species.
It would be interesting to hear comments about the validity of A. jimmadseni, A. atrox, A. amplus, A. lucasi  and contributions to the diagnosis (synapomorphies) of the genus and each one of the species.

---

## Round 0.2 · accepted · Accept

Dear authors,

After reviewing your track changes and response to reviewers, I have decided to 'accept' your manuscript for publication in PeerJ.

You will shortly be contacted by PeerJ production staff regarding your proofs.

The issue of Allosaurus jimmadseni: please liaise with production staff to ensure that your manuscript is published after the one by Chure and Loewen.

Thank you for choosing PeerJ as your publication venue, and I hope you will use us again in the future.